# A Controlled Study of Major Depressive Episodes in Long-Term Childhood, Adolescence, and Young Adult Cancer Survivors (The NOR-CAYACS Study)

**DOI:** 10.3390/cancers13225800

**Published:** 2021-11-18

**Authors:** Alv A. Dahl, Cecilie Essholt Kiserud, Sophie D. Fosså, Jon Håvard Loge, Kristin Valborg Reinertsen, Ellen Ruud, Hanne C. Lie

**Affiliations:** 1National Advisory Unit on Late Effects after Cancer Treatment, Oslo University Hospital, 0424 Oslo, Norway; CKK@ous-hf.no (C.E.K.); sopfos@ous-hf.no (S.D.F.); KVR@ous-hf.no (K.V.R.); h.c.lie@medisin.uio.no (H.C.L.); 2Faculty of Medicine, University of Oslo, 0316 Oslo, Norway; j.h.loge@medisin.uio.no (J.H.L.); Ellen.Ruud@medisin.uio.no (E.R.); 3Department of Behavioral Medicine, University of Oslo, 0316 Oslo, Norway; 4Department of Oncology, Oslo University Hospital, 0406 Oslo, Norway; 5Department of Pediatric Medicine, Oslo University Hospital, Rikshospitalet, 0029 Oslo, Norway

**Keywords:** major depressive episode, PHQ-9, childhood, adolescence and young adult cancer survivors, cancer treatment, cross-sectional study

## Abstract

**Simple Summary:**

A major depressive episode (MDE) is a common mental disorder with profound consequences concerning work ability, comorbidity, and health-related quality of life. Therefore, screening for probable MDE (pMDE) in survivors of childhood and adolescence (CACSs) and young adult cancer (YACSs) survivors is clinically important. This study shows that pMDE is more common among CACSs and YACSs than found in a normative sample using two different definitions of pMDE based on the PHQ-9 screener. pMDE based on a total PHQ-9 score of 10 or more gave higher rates of pMDE than those based on an algorithmic definition. Statistical analyses showed that pMDE according to both definitions was significantly associated with psychosocial factors and self-rated health, while survivor groups, cancer types, and adverse events were not. Screening for pMDE is meaningful in CACSs and YACSs since we have effective treatment methods for pMDE if the condition is identified rather than overlooked.

**Abstract:**

Background: A major depressive episode (MDE) is typically self-rated by screening forms identifying probable MDE (pMDE). This population-based cross-sectional questionnaire study examined the prevalence rates of pMDE identified by the PHQ-9 screener in long-term survivors of childhood and adolescence (CACSs) and young adult cancer (YACSs) and a normative sample (NORMs). Methods: Data from 488 CACSs, 1202 YACSs, and 1453 NORMs were analyzed, and pMDE was defined both by cut-off ≥10 on the total PHQ-9 score and by an algorithm. Results: The prevalence rates of pMDE among CACSs were 21.5%, 16.6% in YACSs, and 9.2% among NORMs using the cut-off definition. With the algorithm, the prevalence rates of pMDE were 8.0% among CACSs, 8.1% among YACSs, and 3.9% among NORMs. Independent of definition, CACSs and YACSs had significantly increased prevalence rates of pMDE compared to NORMs. Psychosocial factors and self-rated health were significantly associated with both definitions of pMDE in multivariable analyses, while survivor groups, cancer types, and adverse events were not. Conclusion: Since pMDE has negative health consequences and is amenable to treatment, healthcare providers should be attentive and screen for pMDE in young cancer survivors. For PHQ-9, the preferred type of definition of pMDE should be determined.

## 1. Introduction

The Global Burden of Disease studies have brought major depressive episode (MDE) to the center stage of international healthcare due to its high prevalence and profound consequences worldwide [1]. MDE is defined by either sadness or loss of interest combined with at least four other symptoms concerning appetite, sleep or energy, guilt, negative thinking, and ideation of death for at least two weeks, with a reduction in previous functioning [2]. Individuals with MDE can experience a single lifetime episode, recurrent episodes, or chronicity, and MDE can sometimes be a manifestation of bipolar disorder [2].

Childhood, adolescent, and young adult cancer survivors (CAYACSs) represent a rapidly growing population known to be at risk of poor mental health. Enduring the traumatic impact of cancer at developmentally vulnerable periods may place CAYACSs at a particularly elevated risk of developing MDE [3]. However, recent reviews and meta-analyses of depression in cancer patients rarely contain studies of CAYACSs [4,5,6].

Several short patient-reported screening instruments of MDE identifying cases in need of clinical examination have been developed. Since the sensitivity and specificity of such screeners are not perfect (see Secton 4), we prefer to use the term probable MDE (pMDE) for cases identified by them. Among such screeners, the Patient Health Questionnaire 9 (PHQ-9) has shown excellent case finding properties and has been recommended for use in oncology [7,8]. Although introduced in 1999, no studies using the PHQ-9 in cancer patients are included in the abovementioned reviews [4,5,6].

Although recommended for oncology, the PHQ-9 has an inherent problem of comprising two ways to identify pMDE, giving significantly different prevalence rates of pMDE. In the literature, PHQ-9-based pMDE is mostly defined either by a cut-off level of the sum score ≥10, or by an algorithm based on the DSM-IV diagnostic criteria for MDE [7,8,9,10]. For example, in a Norwegian population-based study, the prevalence of pMDE based on cut-off ≥10 was 8.1% (95%CI 6.9–9.2%) and that based on the algorithm was 3.2% (95%CI 2.4–4.0%) [11]. The same method-based discrepancy in prevalence rates was also observed in a sample of patients with advanced cancer, with 13.7% pMDE according to the algorithm and 45.3% according to cut-off ≥8 [12]. Adult cancer patients have higher pMDE prevalence rates measured with PHQ-9 cut-off ≥10 than normative samples. For example, a German study found 15.0% among patients and 6.6% in norms [13].

Two previous studies of PHQ-9-based pMDE have been published concerning CAYACSs. Burghardt et al. [14] studied adult long-term survivors of childhood and adolescent cancer (CACSs) at a mean of 28 years after diagnosis. Compared to norms, the odds ratio for pMDE was 3.36, and younger age, but not sex, was significantly associated with possible MDE. However, no prevalence data were given. Geue et al. [15], studying adolescent and young adult cancer survivors (YACSs) <5 years after diagnosis, found that 30% had pMDE using the cut-off definition of ≥9, and that female sex and older age at time of study were significantly associated with higher pMDE rates, but they did not compare their findings with normative data. Further missing from both these studies [14,15] were pMDE prevalence rates based on the PHQ-9 scoring algorithm, comparisons of pMDE between CACSs and YACSs, and of the age and sex distribution of pMDE compared to normative data. However, a previous meta-analysis based on eighteen studies using a variety of depression instruments and definitions found increased risk of depression in CACSs compared to normative data (OR 1.19) [16]. In 2015/2016, a population-based cross-sectional mailed questionnaire study was performed among Norwegian long-term (≥5 years since diagnosis) CAYACSs (the NOR-CAYACS study). They were invited to complete the PHQ-9. In 2015, another population-based study by our group collected data on the PHQ-9 and pMDE from a normative sample (the NORM study) [11]. Since the issue of pMDE is important for the health and quality of life of CAYACSs and relevant for their caretakers, and because the PHQ-9-based definitions of pMDE yield significantly different prevalence rates, we posed the following research questions: (1) Are there significant differences in the prevalence rates of pMDE between CACSs, YACSs, and NORMs based on both the PHQ-9 definitions? (2) What factors are associated with pMDE in bi- and multivariable logistic regression models, and do these associations vary across the two definitions?

## 2. Materials and Methods

### 2.1. Patient Sampling

Since 1953, the Cancer Registry of Norway has, by law, systematically collected notifications on all new cancer cases in the Norwegian population. The Registry contains basic data related to initial diagnosis, disease characteristics, primary treatment, and survival status. Participants eligible for the Norwegian CAYACS study were identified through the Registry. Study inclusion criteria were age ≥18 years at time of survey, diagnosis between 1985 and 2009, and a minimum of 5 years since the initial diagnosis of any childhood and adolescent cancers (excluding central nervous system tumors due to uncertainty about their current cognitive functioning) diagnosed at ages 0–18 years (CACSs); and a selection of cancers diagnosed at ages 19–39 years (YACSs) [13,14].

The YACSs consisted of survivors of breast cancer (stages ≤III), colorectal cancer, non-Hodgkin lymphoma, all leukemias, and a randomly selected subsample of malignant melanomas (960 of 2873). We did not include other common cancer groups such as Hodgkin lymphoma, testicular, and cervical cancer as they were enrolled in concurrent studies at our department at the time of study inclusion [17,18].

A questionnaire was mailed to 5361 CAYACSs, among whom 2104 responded (39% response rate). Based on the returned questionnaires, we excluded survivors with recurrence (*n* = 363) or distant metastases (*n* = 37), due to their supplementary treatment, and questionnaires with no treatment information (*n* = 7) or incomplete PHQ-9 forms (*n* = 7). Thereby, 1690 CAYACS (488 CACSs and 1202 YACSs) entered the analyses.

### 2.2. NORMs

The Bring Dialog Company of Norway randomly drew 6012 subjects, aged 18–80 years, and representative of the general Norwegian population concerning age, sex, and place of residence [11]. Invited persons received a mailed questionnaire packet including the PHQ-9 and three other questionnaires, plus supplementary questions concerning socio-demographics, comorbidities, lifestyle, etc. Non-responders amounted to 3870 persons, and among the 2142 responders who returned the questionnaires (response rate 36%), 1453 aged 18 to 64 years delivered completed PHQ-9 forms.

NORMs as well as CACSs and YACSs were stratified into age groups according to sex for comparative purposes.

### 2.3. Primary Outcome Variables

The PHQ-9 includes the nine diagnostic criteria of MDE according to the DSM-IV classification in a self-rating format [7,19]. The PHQ-9 items are scored as experienced during the last two weeks, and each item is scored from 0 (not at all) to 3 (nearly every day), providing a 0-to-27-dimensional severity score. The items include depressed mood, anhedonia (little interest or pleasure), disturbed sleep, fatigue, eating too much or too little, trouble concentrating, psychomotor retardation or agitation, low self-worth, and suicidal ideation. The internal consistency of the PHQ-9 measured by Cronbach’s coefficient alpha was 0.88.

We identified cases of pMDE by the two recommended definitions [11]: (1) cut-off score ≥10 on the PHQ-9 sum score, and (2) by an algorithm where at least five positive items must be present, of which at least one must be item #1 (anhedonia) or item #2 (depressed mood). Items #1–8 are positive if scored ≥2 (on most days or nearly every day), while item #9 (thoughts about suicide or self-harm) is positive if scored ≥1 (on some days or more). pMDE defined by cut-off or by algorithm were the two primary outcome variables.

### 2.4. Other Instruments

The Fatigue Questionnaire contains a mental (4 items) and a physical fatigue (7 items) subscale, assessing fatigue symptom severity during the past four weeks. Each item is scored from 0 (less/better than usual) to 3 (much more than usual). The total fatigue score is the sum of the subscale scores and ranges from 0 to 33 [20,21]. Cronbach’s alpha was 0.91. To identify cases with chronic fatigue, a dichotomized score for each response alternative (0 = 0, 1 = 0, 2 = 1, 3 = 1) was used with a range of 0 to 11, and chronic fatigue was defined as a dichotomized sum score of ≥4 with a duration of ≥6 months.

The Hospital Anxiety and Depression Scale (HADS) comprises seven items each on the anxiety and depression subscales rated for the last week before the survey. Only the anxiety subscale (HADS-A) was used in this study [22]. The item scores range from 0 (not present) to 3 (highly present), with sum scores ranging from 0 to 21. Probable anxiety disorder was defined by a cut-off subscale score of ≥8. Cronbach’s alpha was 0.83.

The basic personality trait of neuroticism was scored by an abridged version of the Eysenck Personality Questionnaire, with six items concerning long-term characteristics. Each item was scored as present (1) or absent (0). The sum score ranged from 0 to 6 and was dichotomized into high (sum score 3–6) and low neuroticism (sum score 0–2) according to published recommendations [23]. Cronbach’s alpha was 0.77.

### 2.5. Other Measures

Late adverse effects (AEs) were self-reported based on the respondents’ personal experience. Based on the literature [24,25,26], 18 AEs were listed, but we only included fourteen of them that were not covered by other scales or variables: hormonal changes, reduced fertility, dental health problems, cognitive problems, hearing problems, muscular cramps, nerve pains, numbness in hands/feet, second cancer, sexual problems, osteoporosis, lymphedema, radiation injuries, and other problems (to be specified). AEs were only included as present when respondents stated that “I have personal experience with it”. The number of reported AEs was divided into zero (reference), 1–2 and ≥3.

The following somatic diseases were self-reported from a list consisting of cardiovascular diseases, chronic pulmonary diseases, diabetes, kidney diseases, gastrointestinal diseases, rheumatic diseases, arthrosis, stroke, and thyroid diseases. Comorbidity was defined as zero (reference), 1–2 and ≥3 reported diseases. Some of these diseases could also be AEs, but due to lack of data concerning relation to the malignancies and their treatments, they were classified as diseases rather than AEs.

Information on each CAYACS’s cancer type and stage was retrieved from the Registry, while data on primary cancer treatment were self-reported. We defined four treatment groups: limited surgery only (reference, as for localized melanomas), local treatment (surgery and/or radiotherapy), systemic treatment only (chemotherapy and/or hormone therapy), and systemic treatment with surgery and/or radiotherapy.

Age at survey and sex were self-reported. Currently living with a partner was categorized as present (reference) or absent. Level of education was dichotomized into short (≤12 years) and long (>12 years, reference). Current work status had six response alternatives: full- or part-time paid work, being on sick leave, work assessment allowance, disability pension, or others, such as students or homemakers. The responses were dichotomized into “paid work” (full- and part-time work and on sick leave) (reference) versus “not paid work” (the other categories).

Self-rated health had five response alternatives, which were dichotomized into good health (excellent, very good, and good) (reference) versus poor health (moderately poor, poor). Lifestyle variables included obesity defined by body mass index [weight in kilos/(height in meters)^2^] ≥ 30, and current smoking in survivors who reported any number of cigarettes smoked daily at survey.

### 2.6. Statistical Analyses

Between-group comparisons of continuous variables were performed with independent sample t-tests. In the case of skewed distributions, non-parametric tests were used. Between-group comparisons of categorical variables were performed with Fisher’s exact tests. Internal consistencies were given by Cronbach’s coefficient alpha.

We considered sex and age at survey as relevant covariates to be adjusted for when comparing the CACSs and YACSs groups. Adjustments were done by multivariate logistic regression analyses. These adjustments were performed for each of the sex and/or age-relevant independent variables (Table 1). As seen in Table 2, age groups with *n* < 34 were considered too small samples for valid comparisons, as indicated by “not applicable” (NA).

The relationship between independent variables and pMDE, defined by the cut-off and algorithm as dependent variables, were examined with bivariate and multivariable logistic regression analyses. The strength of the associations was given as odds ratios (ORs) with 95% confidence intervals (95%CIs). All variables in the multivariable regression analyses were assessed for multicollinearity.

The *p*-value was set as <0.05, and all tests were two-sided. The software applied was IBM SPSS Statistics version 26 for PC (IBM Corporation, Armonk, NY, USA).

## 3. Results

### 3.1. Responders versus Non-Responders

Both the CAYACSs and NORMs samples contained data on the sex and age of both responders and non-responders. Among responders in both samples, females were significantly over-represented, as were respondents >40 years old among CAYACSs and aged >50 years among NORMSs. The younger age groups were under-represented among respondents in both samples. The mean age was significantly higher among respondents compared to non-respondents.

### 3.2. Sample Characteristics

The CAYACSs sample consisted of 515 (30%) males and 1175 females (70%), with a median age of 31 years (range 0–39) at first cancer diagnosis and 45 years (range 18–64) at survey. Median time since diagnosis was 16 years (range 6–31) (Table 1). The NORMs consisted of 635 (44%) males and 818 (56%) females, with a median age of 49 years (range 18–64) at survey.

Comparing CACSs with YACSs at survey, the CACSs group had significantly lower median age, longer time since diagnosis, and a higher proportion of males (Table 1). Other cancer-related, socio-demographic, somatic and mental health, and lifestyle variables did not show any significant between-group differences when adjusted for sex and age at survey.

### 3.3. pMDE Findings

Based on the cut-off definition, the prevalence rates of pMDE were 21.5% (95%CI 17.9–25.2%) among CACSs, 16.6% (95%CI 14.5–18.7%) among YACSs, and 9.2% (95%CI 7.7–10.7%) among NORMs. Both survivor groups had higher prevalence rates of pMDE than NORMs (*p* < 0.001), while the difference between the survivor groups was non-significant.

According to the algorithm definition, the prevalence rates of pMDE were 8.0% (95%CI 5.6–10.4%) among CACSs, 8.1% (95%CI 6.5–9.6%) among YACSs, and 3.9% (95%CI 2.9–4.8%) among NORMs. Both survivor groups had higher prevalence rates of pMDE than NORMs (*p* < 0.001), while no significant difference was observed between the survivor groups.

In Table 2, the prevalence rates of pMDE based on the cut-off and the algorithm definitions are given separately for sex and age groups among CACSs, YACSs, and NORMs. In the total group of female survivors, the CACSs had a significantly higher prevalence rate of pMDE based on the cut-off than the YACSs (26% versus 18%, *p* = 0.003), while the between-group differences among males were non-significant. Algorithm-defined pMDE showed no significant differences in prevalence rates between CACSs and YACSs according to sex. Comparisons with NORMs according to age groups at survey showed significantly higher prevalence rates of cut-off-defined pMDE in both female and male CACSs aged 30–39 years. Concerning YACSs versus NORMs, females of 50–59 years had significantly higher prevalence rates for both definitions of pMDE, and in the 60–69 years group for pMDE defined by the cut-off. Male YACSs had significantly higher prevalence rates than NORMs on cut-off-defined pMDE in the 40–49 years age group.

### 3.4. Factors Associated with the Two Definitions of pMDE

In the bivariate analyses, all independent variables except time since diagnosis but including CACSs versus YACSs were significantly associated with pMDE defined by the cut-off (Table 3). Similar findings were observed with pMDE defined by the algorithm, except that CACSs versus YACSs, sex, short education, obesity, and daily smoking did not reach significance in the bivariate analyses.

In the multivariable analyses, some independent variables remained significantly associated with both definitions of pMDE: not holding paid work, a probable case of anxiety disorder, chronic fatigue, poor self-rated health, and high neuroticism. In addition, not being in a partnered relationship was significantly associated with pMDE defined by the algorithm. CACSs versus YACSs, cancer treatment, AEs, and comorbidity were not significantly associated with any of the pMDE definitions in the multivariable analyses.

## 4. Discussion

We found that the prevalence of pMDE in CACSs and AYCSs was significantly higher than in NORMs independent of the pMDE definition used, both in the total sample and according to sex. The prevalence rate of pMDE was significantly higher in the total sample of female CACSs versus YACSs, but no such gender difference was found for males.

In the bivariate logistic regression analyses, CACSs versus YACSs was significant for cut-off-defined pMDE, but not for algorithm-defined pMDE. Otherwise, mostly the same independent variables remained significantly associated with both pMDE definitions. In the multivariable analyses, not holding paid work, probable anxiety disorder and chronic fatigue, poor self-rated health, and high neuroticism remained significantly associated with both definitions of pMDE. Mainly psychosocial, and not cancer-related, variables remained significant in the multivariable models.

Due to methodological issues such as follow-up time and lack of prevalence data, our findings are not directly comparable with the previous PHQ-9-based studies of CACSs and YACSs. Geue et al. [16] reported a prevalence of 30% pMDE based on a cut-off score of 9, while we used the cut-off of ten, so our prevalence data cannot be compared. The same problem concerns the study of Burghardt et al. [15], who used a cut-off of ten based on the PHQ-8.

A review of mental disorders or the use of psychiatric medication in YACSs included only three studies and reported an increased risk (OR 1.19) of pMDE compared to cancer-free controls [3]. A systematic review of mental health in YACSs considered eighteen studies based on other screening instruments besides the PHQ-9. The risk for pMDE was OR 1.31 (95%CI 1.12–1.54) based on data from 13,094 YACSs and 7079 norms [16]. Our findings in YACSs are in line with these reviews, showing increased rates of pMDE compared to NORMs. The use of different screening instruments eventually leads to different rates of pMDE.

Other studies of CACSs and CAYACSs have mainly examined mental distress, which is a broad symptom concept covering anxiety, depression, and somatization. Based on mental distress, studies from the Swiss Childhood Cancer Registry, for example, found increased rates of depression in CACSs [27,28] but not in YACSs [29] compared to normative data.

Cancer-related variables did not remain significantly associated with pMDE in our multivariable models. This finding contrasts with a systematic review of 67 CAYACS studies emphasizing the role of cancer treatment and type of cancer for mental distress including depression [30]. Alternatively, current psychosocial status may mediate the effect of cancer-related variables in long-term CAYACSs.

An advantage of studying categorical mental disorders such as pMDE rather than dimensional mental distress is that all major guidelines for diagnosis and treatment concern mental disorders and not distress [31]. In accordance with the high prevalence rate of possible MDE in CAYACSs, there is an increased prescription rate of antidepressants among CAYACSs compared to the general population [32,33]. This is an important clinical point since under-diagnosis of pMDE is a major healthcare problem [34].

We had the opportunity to compare our algorithm-defined point prevalence of pMDE of 7.0% among female and 6.4% among male CAYACs to previously published Norwegian population-based data. These reported the 12-month prevalence of MDE based on an algorithm among females in the capital (Oslo) of 9.7% and in a rural county of 4.5%. Corresponding prevalence, data for males were 4.1% and 3.7% [35,36]. The point prevalence rates of our NORMs were 4.4% for females and 3.1% for males. From these data, we can conclude that male Norwegian CAYACSs have an increased rate of pMDE by any definition compared to normative data for males, while such a conclusion is less obvious for female CAYACSs. However, the prevalence rates found in CAYACSs by the PHQ-9 algorithm approach are mostly in line with the population data. This implies that pMDE rates defined by the PHQ-9 algorithm are somewhat higher compared to population prevalence rates, while cut-off-defined pMDE rates are systematically significantly higher. This discrepancy leads to the question of which PHQ-9-based pMDE definition is the more valid one, since there are practical consequences for CAYACSs if their rates are somewhat or much higher than population rates. This issue could be elucidated by a study of CAYACSs using both PHQ-9 screening definitions and comparing them to findings based on structured interviews for MDE.

Except for sex and age at diagnosis and at survey, we observed surprisingly few differences between the CACSs and YACSs in our sample. These between-group differences were significant in the bivariate analyses when pMDE was defined by the cut-off, but not by the algorithm. For both MDE definitions, we observed significant associations for demographic, somatic and mental health, lifestyle, and cancer-related variables such as treatment modalities and AEs. These findings are in line with the multiple variables model for MDE in cancer patients [37,38].

As indicated, the screening properties of the PHQ-9 differ according to the cut-off level. A sum score cut-off ≥8 has been recommended for oncology [8]. However, since most CAYACSs are controlled by their regular general practitioners, we found a cut-off of ≥10 as recommended for use in general practice (sensitivity and specificity both 0.88 [19]) to be more valid. As discussed by several authors [6,12], the PHQ-9 has several somatic items that could be rated positively by somatic rather than mental problems, and eventually should be omitted in cancer patients. Instead of eliminating these PHQ-9 items, we preferred to apply the higher cut-off score, supported by findings from patients with advanced cancer [12].

Case identification using the PHQ-9 represents a small risk for false positive and false negative diagnoses of MDE. Since we have quite effective pharmacological and psychological treatments for MDE, failure to detect MDE (false negative diagnoses) represents the worst consequence for the patients—particularly in relation to the increased risk of suicide in CAYACSs [39], and especially concerning CACSs [40]. Our findings thereby support the recommendation of screening for MDE in CAYACSs and particularly among female CACSs [5].

Our response rate was 39% among CAYACSs and 36% among NORMSs, which is quite common in population-based questionnaire surveys without any rewards for responding [41,42]. In both samples, there was significant response bias in favor of females, and under-representation among younger and over-representation among older subjects among participants. These biases represent a limitation, and they weaken the generalizability of our findings. However, the potential risk of response bias affecting study outcomes was found to be low in the CAYACS sample [17]. The reliance on self-reporting of cancer treatment, somatic diseases, and some AEs is an obvious limitation of our study. Another limitation is the lack of treatment data for pMDE both among CAYACSs and NORMSs. The pMDE finding abilities of the PHQ-9 vary according to cut-off values and algorithms. In internal comparisons, such as between CACSs and YACSs, this is not a methodological problem. However, in external comparisons, attention should be paid to the pMDE definitions applied. Excluding CNS survivors, known to suffer the greatest risks of late effects, may have lowered prevalence estimates in the CACS group. The cross-sectional design of our sub-study prevents us from drawing causal conclusions.

## 5. Conclusions

We observed increased prevalence rates of pMDE with both definitions among CACSs and YACSs compared to NORMs, but significant differences only in the total female samples of these survivor groups. Given the profound consequences of MDE for work life, economy, risk of suicide, and personal suffering, healthcare providers for CAYACSs should be attentive to the varying but common symptom expression of MDE. If diagnosed, MDE is often amenable to pharmacological and psychological treatments, eventually in combination. The PHQ-9 is a freely available measure that is easy to both administer and score and requires only a couple of minutes for a patient to complete. Given the prevalence of MDE in the cancer survivor population, we recommend the routine use of the PHQ-9, or a comparable screening measure, by providers caring for cancer survivors.

## Figures and Tables

**Table 1 cancers-13-05800-t001:** Characteristics of childhood and adolescent (CACSs) and young adult (YACSs) cancer survivors and their total (CAYACSs) at survey.

Variables	CACSs (*n* = 488)	YACSs (*n* = 1202)	*p*-Value	CAYACSs (*n* = 1690)
Age at first diagnosis (years), median (range)	12 (0–18)	34 (19–39)	<0.001	31(0–39)
Age at survey, median (range)	29 (18–49)	48 (26–64)	<0.001	45 (18–64)
Age at survey, groups	-	-	NA	-
18–29 years	253 (52)	9 (1)		262 (15)
30–39 years	156 (32)	121 (10)		277 (16)
40–49 years	79 (16)	557 (46)		636 (38)
50–59 years	0 (0)	388 (32)		388 (23)
60–64 years	0 (0)	127 (11)		127 (8)
Time since diagnosis, median (range)	20 (5–30)	14 (5–30)	<0.001	16 (5–30)
Types of cancer, *n* (%)	-	283 (24)	NA	
Melanomas	-	472 (39)		283 (17)
Breast	-	129 (11)		472 (28)
Colo-rectal	-	193 (16)		129 (8)
Non-Hodgkin lymphomas	161 (33)	125 (10)		193 (11)
Leukemias	132 (27)	-		286 (17)
Lymphomas	195 (40)	-		132 (8)
Solid tumors				195 (11)
Treatment groups, *n* (%)	-	-	0.94 *	-
Limited surgery only	60 (13)	380 (32)		440 (26)
Local treatment	31 (6)	63 (5)		94 (5)
Systemic treatment only	230 (47)	287 (24)		517 (31)
Systemic + surgery/radiation	167 (34)	472 (9)		639 (38)
Adverse effects, *n* (%)			0.47 *	
None	218 (45)	514 (43)		732 (43)
1–2	168 (34)	297 (25)		465 (28)
≥3	102 (21)	391 (32)		493 (29)
Sex, *n* (%)			<0.001	
Males	202 (41)	313 (26)		515 (30)
Females	286 (59)	889 (74)		1175 (70)
Currently living with a partner, *n* (%)	297 (61)	957 (80)	0.39 *	
Level of education, *n* (%)	-	-	0.81 *	-
Long (>12 years)	270 (55)	701 (59)		971 (58)
Short (≤12 years)	217 (45)	494 (41)		711 (42)
Work status, *n* (%)	376 (77)	1083 (87)	0.86 *	1414 (84)
Paid work	111 (23)	156 (13)		267 (16)
Not paid work				
Comorbidities, *n* (%)	-	-	0.41 *	-
None	322 (66)	722 (60)		1044 (62)
1–2	153 (31)	433 (36)		586 (35)
≥3	13 (3)	47 (4)		60 (3)
Self-rated health, *n* (%)	-	-	0.90 *	-
Good	392 (80)	950 (79)		1342 (79)
Poor	96.(20)	252 (21)		348 (21)
Chronic fatigue cases, *n* (%)	109 (22)	254 (21)	0.56 *	363 (22)
Anxiety cases, *n* (%)	119 (24)	244 (20)	0.75 *	363 (22)
Cut-off MDE cases, *n* (%)	105 (22)	199 (17)	0.98 *	304 (18)
Algorithm MDE cases, *n* (%)	39 (8)	76 (6)	0.42 *	115 (7)
High neuroticism, *n* (%)	212 (44)	410 (34)	0.44 *	622 (37)
Obesity (BMI ≥ 30), *n* (%)	48 (10)	188 (16)	0.32 *	236 (14)
Daily smoking, *n* (%)	30 (6)	138 (12)	0.69 *	168 (10)

NA: Not applicable, *: Adjusted for sex and age at survey.

**Table 2 cancers-13-05800-t002:** Prevalence rates of possible MDE according to the cut-off and the algorithm definitions of PHQ-9 in sex and age groups among CACSs, YACSs, and NORMs.

Sex and Age Groups	CACSs	YACSs	NORMs	*p*-Values		
Females				CACSs vs. YACSs	CACSs vs. NORMs	YACSs vs. NORMs
18–29 years	*n* = 156	*n* = 9	*n* = 68			
PHQ-9 Cut-off, *n* (%)	41 (26)	NA	12 (18)	NA	0.16	NA
PHQ-9 Algorithm, *n* (%)	16 (10)	NA	6 (9)	NA	0.74	NA
30–39 years	*n* = 84	*n* = 88	*n* = 120			
PHQ-9 Cut-off, *n* (%)	23 (27)	17 (19)	13 (11)	0.21	0.002	0.09
PHQ-9 Algorithm, *n* (%)	8 (10)	7 (8)	9 (8)	0.72	0.61	0.9
40–49 years	*n* = 46	*n* = 416	*n*= 228			
PHQ-9 Cut-off, *n* (%)	10 (22)	79 (19)	31 (14)	0.65	0.15	0.08
PHQ-9 Algorithm, *n* (%)	2 (4)	23 (6)	10 (4)	1	1	0.52
50–59 years	*n* = 0	*n* = 283	*n* = 260			
PHQ-9 Cut-off, *n* (%)	NA	45 (16)	24 (9)	NA	NA	0.021
PHQ-9 Algorithm, *n* (%)	NA	18 (6)	7 (3)	NA	NA	0.042
≥60 years	*n* = 0	*n* = 93	*n* = 142			
PHQ-9 Cut-off, *n* (%)	NA	13 (14)	9 (6)	NA	NA	0.049
PHQ-9 Algorithm, *n* (%)	NA	6 (7)	4 (3)	NA	NA	0.2
Total	*n* = 286	*n* = 889	*n* = 819/817			
PHQ-9 Cut-off, *n* (%)	74 (26)	158 (18)	89 (11)	0.003	<0.001	<0.001
PHQ-9 Algorithm, *n* (%)	26 (9)	56 (6)	36 (4)	0.11	0.003	0.08
Males				CACSs vs. YACSs	CACSs vs. NORMs	YACSs vs. NORMs
18–29 years	*n* = 97	*n* = 0	*n* = 36			
PHQ-9 Cut-off, *n* (%)	13 (13)	NA	4 (11)	NA	1	NA
PHQ-9 Algorithm, *n* (%)	6 (6)	NA	1 (3)	NA	0.67	NA
30–39 years	*n* = 72	*n* = 33	N = 816 (7)			
PHQ-9 Cut-off, *n* (%)	14 (19)	4 (12)	3 (4)	0.36	0.027	0.47
PHQ-9 Algorithm, *n* (%)	5 (7)	1 (3)		0.66	0.48	1
40–49 years	*n* = 33	*n* = 141	*n*= 166			
PHQ-9 Cut-off, *n* (%)	4 (12)	23 (16)	14 (8)	0.55	0.51	0.035
PHQ-9 Algorithm, *n* (%)	2 (6)	12 (9)	6 (4)	1	0.62	0.07
50–59 years	*n* = 0	*n* = 105	*n*= 216			
PHQ-9 Cut-off, *n* (%)	NA	11 (11)	13 (6)	NA	NA	0.15
PHQ-9 Algorithm, *n* (%)	NA	5 (5)	6 (3)	NA	NA	0.35
≥60 years	*n* = 0	*n* = 34	*n* = 136			
PHQ-9 Cut-off, *n* (%)	NA	3 (9)	8 (6)	NA	NA	0.46
PHQ-9 Algorithm, *n* (%)	NA	2 (6)	4 (3)	NA	NA	0.34
Total	*n* = 202	*n* = 313	*n* = 638/635			
PHQ-9 Cut-off, *n* (%)	31 (15)	41 (13)	45 (7)	0.47	<0.001	0.002
PHQ-9 Algorithm, *n* (%)	13 (6)	20 (6)	20 (3)	0.98	0.035	0.019

NA = Not applicable due to small numbers (*n* < 34).

**Table 3 cancers-13-05800-t003:** Bivariate and multivariate logistic regression analyses of independent variables and possible MDE defined by the cut-off and the algorithm definitions.

Variables	MDE Defined by PHQ-9 Cut-Off Score ≥10						MDE Defined by Algorithm					
	Bivariate Analyses			Multivariable Analysis			Bivariate Analysis			Multivariable		
	OR	95%CI	*p*-value	OR	95%CI	*p*-value	OR	95%CI	*p*-value	OR	95%CI	*p*-value
Age at survey	0.98	0.97–0.99	<0.001	0.99	0.97–1.01	0.37	0.98	0.97–1.00	0.039	1.01	0.99–1.03	0.32
Time since diagnosis	0.98	0.97–1.00	0.064				0.98	0.95–1.01	0.15			
AYCSs (CACSs = reference)	0.72	0.56–0.94	0.016	0.89	0.51–1.57	0.69	0.78	0.52–1.16	0.218	-	-	-
Treatment groups	-	-	-	-	-	0.94		-	-	-	-	0.15
Surgery only (reference)	1	-	-	1	-	-	1	-	-	1	-	-
Local treatment												
Systemic treatment only	1.72	0.93–3.18 1.45–3.05	0.087	1.09	0.44–2.71	0.86	1.71	0.60–4.87	0.315	0.93	0.23–3.72	0.92
Systemic + other treatments	2.11		<0.001	0.99	0.55–1.77	0.97	3.19	1.73–5.85	<0.001	1.73	0.78–3.85	0.18
		1.60–3.26										
	2.28		<0.001	0.92	0.52–1.64	0.79	2.42	1.31–4.44	0.005	0.99	0.43–2.26	0.97
Comorbidities	-	-	-	-	-	0.84	-	-	-	-	-	0.34
None (reference)	1	-	-	1	-	-	1	-	-	1.00 0.73 1.23	-	-
1–2	1.86	1.44–2.41	<0.001	1.11	0.76–1.62	0.58	1.42	0.95–2.12	0.084		0.45–1.21	0.22
3+	3.05	1.74–5.37	<0.001	0.98	0.44–2.23	0.97	3.34	1.61–6.92	0.001		0.47–3.23	0.67
Male (female reference)	0.66	0.50–0.88	0.005	0.72	0.47–1.09	0.12	0.91	0.60–1.39	0.67	-	-	-
Not living with a partner	1.85	1.42–2.41	<0.001	1.36	0.92–1.99	0.12	2.22	1.51–3.27	<0.001	1.86	1.15–3.00	0.012
Short basic education	1.6	1.25–2.06	<0.001	1.42	098–2.04	0.06	1.37	0.94–3.00	0.1	-	-	-
Not in paid work	3.76	2.90–4.88	<0.001	1.89	1.29–2.76	0.001	3.33	2.72–4.89	<0.001	1.72	1.06–2.80	0.029
Probable anxiety case	14.11	10.58–18.80	<0.001	5.26	3.60–7.67	<0.001	21.72	13.29–35.48	<0.001	7.71	4.38–13.57	<0.001
Obesity	1.83	1.33–2.52	<0.001	1.52	0.96–2.41	0.08	1.6	0.99–2.58	0.06	-	-	-
Daily smoking	1.78	1.24–2.57	0.002	1.05	0.62–1.79	0.85	1.39	0.79–2.46	0.25	-	-	-
Chronic fatigue	14.52	10.64–19.84	<0.001	5.19	3.61–7.45	<0.001	10.43	6.87–15.84	<0.001	3.66	2.21–6.04	<0.001
Poor self-rated health	9.32	7.07–12.30	<0.001	3.06	2.09–4.47	<0.001	6.97	4.70–10.34	<0.001	1.97	1.20–3.21	0.007
High neuroticism	15.6	11.12–21.88	<0.001	4.02	2.64–6.13	<0.001	10.02	6.69–15.02	<0.001	2.67	1.33–5.36	0.006
Adverse effects	-	-	-	-	-	0.13	-	-	-	-	-	0.24
Zero (reference)	1	-	-	1	-	-	1	-	-	1	-	-
1–2	2.26	1.62–3.15	<0.001	1.63	1.00–2.66	0.05	2.98	1.76–5.04	<0.001	1.74	0.88–3.44	0.11
≥3	3.51	2.57–4.79	<0.001	1.52	0.91–2.54	0.11	3.56	2.14–5.90	<0.001	1.29	0.63–2.65	0.49

## Data Availability

The dataset is available from H.C.L., the project leader.

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
