# Peer review of "A Controlled Study of Major Depressive Episodes in Long-Term Childhood, Adolescence, and Young Adult Cancer Survivors (The NOR-CAYACS Study)"

_cancers, 2021, doi:10.3390/cancers13225800_

Round 1

Reviewer 1 Report

The mansucript by Dahl et a. entitled 'A controlled study of major depressive episodes in long-term childhood, adolescence, and young adult cancer survivors (The NOR-CAYACS study) reports on probable major depressive episodes (pMDE) in cancer survivors. Given that untreated mood disorders can significantly lower quality of life and general health, this is an important study to ensure optimal treatment of cancer survivors.

However, I have some concerns regarding the study design.

On p.3, l. 114 it is stated that the number of YACS included in the analysis is 960 out of 2,873, however in table 1, the number of YACS is 1,202. The number of CACS is not stated in the method section (p.3). I highly recommend specifying both numbers in the method section in the two patient cohorts separately.

Self-reported health in cancer survivors was considered when comparing YACS and CACS. However, was general health also considered when comparing NORMS with cancer survivors?

In addition, it seems that current medication was not considered in the two patient groups. Given that medication can have an impact on major depressive episodes, this should be discussed.

Editorial:

p3, l. 114: Missing full-stop after (960 out of 2,873)

Author Response

Reviewer 1

Our comments are given in italics.

Open Review

English language and style

( ) Extensive editing of English language and style required
( ) Moderate English changes required
(x) English language and style are fine/minor spell check required
( ) I don't feel qualified to judge about the English language and style

We are thankful  for the Reviewer’s judgement that our English language only requires “minor spell check.

                                                                      Yes

Can be improved

Must be improved

Not applicable

Does the introduction provide sufficient background and include all relevant references?

(x)

( )

( )

( )

Is the research design appropriate?

( )

(x)

( )

( )

Are the methods adequately described?

( )

( )

(x)

( )

Are the results clearly presented?

(x)

( )

( )

( )

Are the conclusions supported by the results?

( )

(x)

( )

( )

We register that the Reviewer requires improvements of our Methods description.

Comments and Suggestions for Authors

The mansucript by Dahl et a. entitled 'A controlled study of major depressive episodes in long-term childhood, adolescence, and young adult cancer survivors (The NOR-CAYACS study) reports on probable major depressive episodes (pMDE) in cancer survivors. Given that untreated mood disorders can significantly lower quality of life and general health, this is an important study to ensure optimal treatment of cancer survivors.

However, I have some concerns regarding the study design.

On p.3, l. 114 it is stated that the number of YACS included in the analysis is 960 out of 2,873, however in table 1, the number of YACS is 1,202. The number of CACS is not stated in the method section (p.3). I highly recommend specifying both numbers in the method section in the two patient cohorts separately.

We make apologies for this confusion. As we state, the 960 is a “randomly selected subsample of malignant melanomas (960 of a total of 2,873)” not the number of YACS. Table 1 states that 283 of them were included among the 1,202 YACS across diagnostic groups. We hope this is clear in the text as stands.

In line 121 we state: “Thereby 1,690 CAYACS (488 CACSs and 1,202 YACSs) entered the analyses.” We thereby consider that we have fulfilled the Reviewer’s wish for “specifying both numbers in the method section in the two patient cohorts separately.”

Self-reported health in cancer survivors was considered when comparing YACS and CACS. However, was general health also considered when comparing NORMS with cancer survivors?

The Reviewer raises an interesting issue here. The question on “self-rated health” used in the CAYACS questionnaire was not included in the NORMS questionnaire. However, the NORMS questionnaire includes the SF-36 form, where one subscale is “General health”. Unfortunately. due to different wording and scoring of these issues, we were not able to consider general health when comparing NORMS with cancer survivors. We, therefore, did not change the manuscript concerning this comment by the Reviewer.

In addition, it seems that current medication was not considered in the two patient groups. Given that medication can have an impact on major depressive episodes, this should be discussed.

We fully agree with Reviewer on this point, and we have added a comment on this issue under Limitation in the Discussion section of the revised manuscript.

Editorial:

p3, l. 114: Missing full-stop after (960 out of 2,873)

We have corrected that point in the revised manuscript.

Reviewer 2 Report

The literature on depression in childhood and AYA cancer survivors, as is typical for this population, is mostly based on studies with modest sample sizes. The sample size of the current study and the fact that the survivors were selected from a cancer registry (despite the relatively low response rate) make this a valuable contribution. I found that the study is generally well justified, the methodology well described and appropriate, and the discussion balanced.

I agree with the authors’ approach: having in mind that they want to compare depressive episode prevalence in survivors with that of the general population, it is reasonable to apply the same cut-off to both. I also would not find it very justifiable to remove the somatic items because, due to cancer or not, they contribute to depression.

I also agree (and it was very surprising to me when I found out at the time) that most depression prevalence studies in cancer survivors in general and using screeners report estimates from other instruments but not the PHQ-9, which is the one instrument based on the DSM’s diagnostic criteria. Here I would like to draw the authors’ attention to a very similar recent study that used the PHQ-9 albeit in adult cancer survivors: 10.3390/cancers13133368

Below I include some specific questions or comments regarding the results and discussion:

In the methods section, it is not clear whether the adverse effects and somatic diseases were chosen from a list or reported in open ended questions and then categorized by researchers, please clarify.

“Current paired relation” sounds awkward to me in English, change to “Currently in a romantic relationship”?

The survivor and the norm populations differ on basic demographics such as age and sex, which can contribute to differences in depression prevalence. Comparing prevalence in the different age groups as the authors did partially solves this problem. An alternative approach would be to draw controls matched on age and gender from the general normative sample and only use this sample of controls for the estimation of prevalence and comparisons (as in the paper previously mentioned: 10.3390/cancers13133368). However, looking at Table 2 I am not sure this would be feasible, I mentioned it anyway as an approach to consider.

I am not sure if something has moved in Table 3 or I am misunderstanding it. Consider for example the section on age and time since diagnosis, there is only one coefficient visible for the multivariable model. Shouldn’t there be two?

In the multivariable models mostly the psychosocial factors explain depression scores. Cancer related variables and comorbidities were, however, significant in univariate analysis. In my opinion, this supports the alternative explanation that the authors propose in the discussion, that the effect of cancer and its treatment on depression is probably at least partially mediated by its influence on psychosocial factors. We know that cancer (treatment) affects childhood and AYA survivors globally, including physical, mental, and social outcomes, and the current understanding of the influence of cancer and other chronic diseases on mental health outcomes supports that these factors are strongly interrelated. It is unlikely that cancer (treatment) contributes to an increased depression risk only by biological mechanisms.

Would it be worth mentioning in the discussion how the prevalence established in the current study compares to the prevalence found in similar studies but that used other instruments? In case there are studies that would provide a reasonable comparison, it might be interesting to discuss some differences between the results obtained with different instruments.

Thank you for the opportunity to review this article, I hope my comments are useful to improve the manuscript.

Author Response

Reviewer 2                                                                                                                                       Our comments are given in italics.

Open Review

English language and style

( ) Extensive editing of English language and style required
( ) Moderate English changes required
(x) English language and style are fine/minor spell check required
( ) I don't feel qualified to judge about the English language and style

We are thankful  for the Reviewer’s judgement that our English language only requires “minor spell check.

Yes

Can be improved

Must be improved

Not applicable

Does the introduction provide sufficient background and include all relevant references?

(x)

( )

( )

( )

Is the research design appropriate?

(x)

( )

( )

( )

Are the methods adequately described?

( )

(x)

( )

( )

Are the results clearly presented?

( )

(x)

( )

( )

Are the conclusions supported by the results?

(x)

( )

( )

( )

Comments and Suggestions for Authors

The literature on depression in childhood and AYA cancer survivors, as is typical for this population, is mostly based on studies with modest sample sizes. The sample size of the current study and the fact that the survivors were selected from a cancer registry (despite the relatively low response rate) make this a valuable contribution. I found that the study is generally well justified, the methodology well described and appropriate, and the discussion balanced.

I agree with the authors’ approach: having in mind that they want to compare depressive episode prevalence in survivors with that of the general population, it is reasonable to apply the same cut-off to both. I also would not find it very justifiable to remove the somatic items because, due to cancer or not, they contribute to depression.

We are thankful for the Reviewer’s opinions on these matters.

I also agree (and it was very surprising to me when I found out at the time) that most depression prevalence studies in cancer survivors in general and using screeners report estimates from other instruments but not the PHQ-9, which is the one instrument based on the DSM’s diagnostic criteria. Here I would like to draw the authors’ attention to a very similar recent study that used the PHQ-9 albeit in adult cancer survivors: 10.3390/cancers13133368

Below I include some specific questions or comments regarding the results and discussion:

In the methods section, it is not clear whether the adverse effects and somatic diseases were chosen from a list or reported in open ended questions and then categorized by researchers, please clarify.

We understand the Reviewer’s need for specification on these issues. As stated in line 168-169: “18 AEs were listed, but we only included 14 of them that were not covered by other scales or variables” So AEs were based on a list made from references #24-26, and we did not change the revised manuscript on that issue.                                                                           The somatic diseases were also chosen from a list, and we have added that information in the revised manuscript.

“Current paired relation” sounds awkward to me in English, change to “Currently in a romantic relationship”?                                                                                                                We are somewhat in doubt concerning this point by the Reviewer due to the following facts. We have asked if the CAYACSs are married or living with another person  and several alternatives, and these have been dichotomized. The degree of romanticism has not been asked for. To compromise with the Reviewer, we have changed the term to “Living with a partner”, and we hope that will be approved.                                                                                                   

The survivor and the norm populations differ on basic demographics such as age and sex, which can contribute to differences in depression prevalence. Comparing prevalence in the different age groups as the authors did partially solves this problem. An alternative approach would be to draw controls matched on age and gender from the general normative sample and only use this sample of controls for the estimation of prevalence and comparisons (as in the paper previously mentioned: 10.3390/cancers13133368). However, looking at Table 2 I am not sure this would be feasible, I mentioned it anyway as an approach to consider.                 We are thankful for the Reviewer’s suggestion on this issue. As will be seen from Table 2, the NORMSs are under-represented in the 18-29 years age group for both genders, which preclude drawing of controls matched on age and gender. This comment has not led to any changes in the revised manuscript.

I am not sure if something has moved in Table 3 or I am misunderstanding it. Consider for example the section on age and time since diagnosis, there is only one coefficient visible for the multivariable model. Shouldn’t there be two?                                                                                   We thank the reviewer for pointing this error out. In fact, these should only be one, but as correctly indicated by the Reviewer the multivariable data concerns the Age at survey and should be on that line. This has moved in the process of transferal of the original manuscript and should be corrected by the Editorial office.

In the multivariable models mostly the psychosocial factors explain depression scores. Cancer related variables and comorbidities were, however, significant in univariate analysis. In my opinion, this supports the alternative explanation that the authors propose in the discussion, that the effect of cancer and its treatment on depression is probably at least partially mediated by its influence on psychosocial factors. We know that cancer (treatment) affects childhood and AYA survivors globally, including physical, mental, and social outcomes, and the current understanding of the influence of cancer and other chronic diseases on mental health outcomes supports that these factors are strongly interrelated. It is unlikely that cancer (treatment) contributes to an increased depression risk only by biological mechanisms.        The authors are fully in line with the Reviewer on this point. However, we are in doubt if the Reviewer suggests that the manuscript needs revision on this point (lines #309-313)?.

Would it be worth mentioning in the discussion how the prevalence established in the current study compares to the prevalence found in similar studies but that used other instruments? In case there are studies that would provide a reasonable comparison, it might be interesting to discuss some differences between the results obtained with different instruments.                              We see the point of the Reviewer here, but we are afraid of raising another and lengthy issue. We therefore only added a sentence (line #304) indicating the issue without going further into it.

Thank you for the opportunity to review this article, I hope my comments are useful to improve the manuscript.                                                                                   We thank the Reviewer for thoughtful and helpful comments.